# Trabecular Titanium Architecture Drives Human Mesenchymal Stem Cell Proliferation and Bone Differentiation

**DOI:** 10.3390/ijms26136354

**Published:** 2025-07-01

**Authors:** Laura Caliogna, Micaela Berni, Giulia Gastaldi, Federico Alberto Grassi, Eugenio Jannelli, Mario Mosconi, Elisa Salatin, Silvia Burelli, Riccardo Toninato, Michele Pressacco, Gianluigi Pasta

**Affiliations:** 1Orthopedics and Traumatology Clinic, IRCCS Policlinico San Matteo Foundation, 27100 Pavia, Italy; l.caliogna@smatteo.pv.it (L.C.); federico.grassi@unipv.it (F.A.G.); eugenio.jannelli@unipv.it (E.J.); mario.mosconi@unipv.it (M.M.); gianluigipasta@yahoo.it (G.P.); 2Department of Molecular Medicine, University of Pavia, 27100 Pavia, Italy; giulia.gastaldi@unipv.it; 3Centre for Health Technologies, University of Pavia, 27100 Pavia, Italy; 4Department of Clinical, Surgical, Diagnostic and Pediatric Sciences, University of Pavia, 27100 Pavia, Italy; 5Limacorporate Spa, 33030 Villanova di San Daniele del Friuli, Italy; elisa.salatin@enovis.com (E.S.); silvia.burelli@enovis.com (S.B.); riccardo.toninato@enovis.com (R.T.); michele.pressacco@enovis.com (M.P.)

**Keywords:** trabecular titanium, bone differentiation, osteoinduction, 3D printing, additive manufacturing, human adipose-derived mesenchymal stem cells

## Abstract

The aim of this *in vitro* study is to investigate the adhesion, proliferation, and differentiation of human adipose-derived mesenchymal stem cells (hASC) on Trabecular Titanium scaffolds manufactured with different manufacturing processes (EBM and SLM). The *in vitro* adhesion and proliferation of hASC on titanium scaffolds with WST assays have been carried out. The comparison of the gene expression profiles of typical bone genes (*Alp*, *Bglap*, *Col1a1*, and *Osx*) through real-time PCR assays and the evaluation of extracellular matrix composition with immunofluorescence and SEM analysis have been performed. In addition, the possible osteoinductive properties of the two scaffolds have been investigated through real-time PCR and ALP assays. Data showed that Trabecular Titanium supports human adipose-derived mesenchymal stem cell colonization and induces differentiation in bone with the deposition of the abundant extracellular mineralized matrix regardless of the manufacturing process, proving that the micro- and macro-design features are the key factors responsible for the osteoinduction behavior. These features can only be achieved through tailored 3D printing process parameters.

## 1. Introduction

The quest to enhance bone regeneration and integration with implants has driven extensive research into the design of biomaterials that facilitate natural bone growth, particularly within the constrained porosities of cellular solids and metal foams. It is well-established that successful bone ingrowth depends on several key factors: the appropriate surface roughness of the material, its biocompatibility, and the optimal size and distribution of its pores [1,2]. These attributes significantly influence bone formation both *in vitro* and *in vivo*, as they contribute to the ability to support cell colonization, migration, and the formation of a functional bone matrix [3]. Porous biomaterials offer the advantage of a high surface area at the implant–bone interface, providing essential biological anchorage to surrounding tissues. Cells are particularly sensitive to the topography and chemistry of the surface they interact with, making precise control of roughness at multiple scales (macroscale, microscale, and nanoscale) crucial for promoting osteointegration [1].

Among the innovations in porous biomaterials, Trabecular Titanium (TT) stands out as a highly porous structure that mimics the morphology of the trabecular bone with an elastic modulus tailored to match that of human bone, which allows for a uniform load transfer, avoiding stress-shielding phenomena [4,5,6]. The TT three-dimensional structure (EP2164428B1, US8454705) is derived from the replica of a native unit cell and manufactured in this study using Electron-Beam Melting (EBM) and Selective Laser Melting (SLM) techniques. A schematic illustration (Figure 1) of TT manufacturing processes is shown below. The fully interconnected porous architecture exhibits an average pore size of 640 ± 100 µm, a dimension specifically selected to support bone ingrowth and vascularization [1].

The overall porosity is approximately 65%, closely resembling that of natural cancellous bone [4,5]. Previous studies on mechanical characterization investigated TT adhesion and compression properties in both EBM and SLM technologies [5,6,7]. The structure’s inherent macro-scale surface roughness provides a high coefficient of friction against bone, promoting the primary mechanical fixation of the implant. The presence and functional role of surface irregularities have been indirectly assessed through friction testing. This ability to precisely control features and properties makes TT a proper candidate for orthopedic applications, ensuring primary stability and biological fixation of the implant [8,9,10,11].

Due to the maturity of the technology, TT was historically developed using Electron-Beam Melting (EBM), a 3D printing technology capable of processing reactive materials such as titanium alloys and realizing complex porous structures with proven mechanical properties [12,13]. However, in recent years, Selective Laser Melting (SLM) has been strongly optimized and emerged as another viable method for this purpose. Both EBM and SLM are powder-bed fusion 3D printing techniques that involve the selective melting of metal powder layers to build a three-dimensional structure based on a digital model. Despite their shared principles, these methods differ in some key aspects, including energy source, operating environment, and temperature, as well as the post-processing steps required. EBM uses a high-energy electron beam and operates in a high-vacuum chamber to prevent beam scattering by air molecules. SLM, on the other hand, employs a high-power laser and operates in an inert gas atmosphere (usually argon) to prevent oxidation during the melting process. Despite these differences, specific optimization of both process parameters and post-processing has enabled SLM to successfully produce TT structures with mechanical, frictional, and chemical properties comparable to those achieved with EBM [7,9]. Trabecular Titanium manufactured via EBM has been shown to produce a microenvironment that can stimulate mesenchymal stem cells obtained from human adipose tissue or bone marrow to adhere, proliferate, and differentiate towards osteoblastic lineage, producing a calcified extracellular matrix, which also occurs in the absence of osteogenic factors [14,15,16].

This recent technology advancement raises questions about whether differences in manufacturing processes might influence the biological behavior of TT structures. This study investigates the osteogenic potential of TT scaffolds produced via different 3DP technologies by evaluating the behavior of human adipose-derived stem cells (hASCs) *in vitro*. Specifically, it evaluates the adhesion, proliferation, and osteogenic differentiation of hASCs on TT scaffolds manufactured via EBM and SLM, analyzing the expression of typical osteogenic genes, such as *COL1A1*, *OSX*, *ALP*, and *BGLAP*, as well as alkaline phosphatase (ALP) activity. The production of the extracellular matrix is investigated using immunofluorescence and SEM analysis.

## 2. Results

### 2.1. Cell Proliferation and Osteogenic Differentiation

#### 2.1.1. Adhesion and Proliferation Assay

The proliferation of hASCs was evaluated on 40 scaffolds (20 TT EBM and 20 TT SLM) using the WST test at different time points: 24 h, 7 days, 14 days, 21 days, and 28 days. The results showed that about 90% of hASC adhered to the scaffold surface and were able to proliferate. Both the adhesion result and the growth curves were similar between the two scaffold sets. The cells exhibited comparable trends in both growth medium (GM, 10 scaffolds per set) and osteogenic differentiation medium (OM, 10 scaffolds per set) (Figure 2). The results did not show a significant statistical difference.

#### 2.1.2. Alkaline Phosphatase Activity (ALP)

To evaluate osteogenic differentiation, alkaline phosphatase (ALP) activity was analyzed in cells seeded on 60 TT EBM scaffolds and 60 TT SLM scaffolds at 7, 14, and 21 days.

In GM medium, ALP activity showed no significant differences between TT EBM and TT SLM scaffolds. In OM medium, cells on TT EBM scaffolds exhibited higher ALP activity compared to cells grown on TT SLM scaffolds, but the differences were not statistically significant.

Overall, ALP activity trends were comparable between the two scaffold types (Figure 3).

ALP activity was also evaluated in 60 monolayer cultures (5 per scaffold type per time point) to determine whether cells growing on scaffolds release a secretome with osteoinductive effect. The analysis was conducted 7, 14, and 21 days after osteogenic differentiation. The results showed an increased ALP activity in hASCs cultured in conditioned medium (CM) compared to GM. No differences in ALP protein expression were observed between scaffold types (Figure 4). These data suggest that both scaffold sets induce stem cell differentiation in the absence of osteogenic factors.

#### 2.1.3. mRNA Expression

qRT-PCR analysis was performed on 40 scaffolds (20 TT EBM and 20 TT SLM) and 40 monolayer cultures. The specificity of the PCR reaction is confirmed by the presence of a single peak corresponding to a single amplicon in the melt curve plot. Gene expression levels of *Alp*, *Col1a1*, *Osx*, and *Bglap* were expressed as fold change relative to the gene expression of hASC seeded on a monolayer and cultured in the absence of osteogenic medium (control). The results revealed similar gene expression patterns between cells seeded on TT SLM scaffolds (10 scaffolds for GM and 10 scaffolds for OM) and TT EBM scaffolds (10 scaffolds for GM and 10 scaffolds for OM). No significant differences were observed between scaffolds cultured in GM or OM (Figure 5).

The analysis conducted on monolayer (10 for GM, 10 for TT EBM CM, 10 for TT SLM CM, and 10 for OM) revealed an increase in bone gene expression when cells were grown in CM compared to GM. Specifically, there was a statistically significant increase in *col1a1* and *osx* expression in cells treated with CM, while *alp* and *bglap* expression showed an increase but not statistically significant. No significant differences in gene expression were observed between hASCs grown in EBM or SLM conditioned medium (Figure 6).

#### 2.1.4. Immunofluorescence Assay

An immunofluorescence assay was conducted on 10 scaffolds (5 TT EBM and 5 TT SLM) to evaluate the presence of collagen in the extracellular matrix on different scaffolds. Both TT SLM and TT EBM exhibited a deposition of collagen, evidenced by the red staining (Figure 7). These results showed that the cells were able to differentiate in an osteogenic direction both on SLM and EBM scaffolds and did not show a significant difference in bone matrix deposition on the structure’s surface.

#### 2.1.5. SEM Analysis

SEM analysis conducted on 10 scaffolds (5 per type) showed that both TT EBM and TT SLM scaffolds in GM were colonized by cells (Figure 8). To visualize cells and the matrix, images were captured using a backscattered electron detector (BSE). The metallic scaffold appeared white (inorganic zone), while the matrix and the cells appeared dark (organic zone) (Figure 8B,D).

SEM images on 12 scaffolds (6 per type) showed an abundant matrix secreted by hASC cultured in the presence of osteogenic factors and differentiated into osteoblasts on TT EBM and TT SLM scaffolds (Figure 9).

At 21 days post-differentiation, bone matrix deposition was observed covering the scaffold’s surface. The matrix was homogeneous and similar on both TT EBM and TT SLM scaffolds. EDS analysis revealed the presence of calcium (Ca) and phosphate (P), confirming that the matrix was ossified (Figure 10). The analysis was repeated at 40 days post-differentiation and confirmed the results obtained at 21 days. The data showed that hASC grown on EBM and SLM scaffolds did not exhibit significant differences in the deposition and composition of the bone matrix.

## 3. Discussion

Trabecular Titanium (TT) is a biomaterial designed to mimic the trabecular bone to enhance the integration with native bone tissue, making TT an optimal choice for orthopedic implants, such as acetabular cups. The TT structure is engineered to combine biomechanical properties and biological fixation. Previous studies have demonstrated that its specific surface roughness enhances the microenvironment for cellular adhesion, proliferation, and differentiation, making it particularly suitable for applications in bone regeneration [14,15,16].

Building on these findings, this study sought to investigate whether the manufacturing process of TT influences its biological performance when micro and macro surface features are guaranteed. After a deep and robust refinement of the SLM process and related post-processing steps to faithfully replicate the characteristic porosity, roughness performances, and structural integrity of EBM-manufactured TT [6,7,9], the results confirmed that the biological response of cells is driven by the intrinsic properties of the structure itself, rather than the specific manufacturing technology employed. This highlights that the key factors promoting osteogenesis are inherent to TT’s structural and surface characteristics, underscoring its robustness and versatility for orthopedic applications [17].

To evaluate whether the scaffold manufacturing process (EBM and SLM) interfered with the biocompatibility of the scaffolds, a standard protocol for cell seeding of the biomaterials was used. This protocol included the monitoring of cell adhesion and proliferation both in growth medium and in the presence of osteogenic factors, the evaluation of gene expression of genes typical of bone differentiation in real time, and the evaluation of collagen deposition, a protein essential for good mineralization of the bone matrix through SEM and immuno assay. Several literature data confirm that the protocol used in this article, with small modifications depending on the material needs, is commonly accepted and used to test different types of biomaterials, including titanium and various polymers such as PEEK, PGLA, and silk fibroin [18,19,20,21].

The results obtained in this study confirm this hypothesis. The data showed that hASCs can adhere well to—and proliferate on—both TT EBM and TT SLM scaffolds, regardless of the manufacturing technology. The cells grew effectively on both scaffolds and exhibited good proliferation, confirming that each scaffold is an excellent biocompatible material.

In osteogenic conditions, the cells differentiated well on both TT EBM and TT SLM scaffolds, with each scaffold demonstrating a comparable mineralization rate, as confirmed by ALP enzymatic assay. Additionally, the analysis showed a statistically significant increase in ALP activity in cells grown with CM, obtained from cultures of EBM and SLM scaffolds, compared to cells grown on a monolayer with GM. These data confirmed that TT scaffolds have good osteoinductive properties as previously reported [15]. To assess the cells’ ability to differentiate into bone, a real-time PCR assay was performed to evaluate the expression of typical bone genes (*col1a1*, *alp*, *osx*, and *bglap*) [22]. The analysis was conducted on cells grown on TT EBM and TT SLM scaffolds, as well as on cells grown in a monolayer with GM, CM, and OM medium.

*Alp* (alkaline phosphatase) is considered both an early and late marker of osteogenic differentiation, playing a key role in bone mineralization and the initial phase of differentiation [23]. *Col1a1* (Collagen alpha-1-chain) is typically expressed in osteogenic differentiation and mineralization, while in bone remodeling, *Bglap* (Osteocalcin) is usually involved because it is associated with energy metabolism in osteoblasts and calcium binding. *Osx* (Transcription factor Sp7) inhibits osteoblasts’ mature markers like *Bglap* and *Col1a1* and regulates the osteoblast differentiation [24,25,26].

Cells grown on TT EBM and TT SLM scaffolds showed a similar bone gene expression, confirming that the manufacturing process did not interfere with cell differentiation.

In monolayer, the cells grew in CM, highlighting a significant increase in *col1a1* and *osx* expression and an increasing trend of *alp* and *bglap* expression, showing the two scaffolds had osteoinductive properties.

This property is probably due to both the characteristics of the scaffold, since the three-dimensional structure mimics trabecular bone, and to the mechanical properties of the scaffold, especially the elastic characteristics that are similar to spongy bone.

The hASC grown on the three-dimensional scaffolds would seem to release factors into the medium that induce differentiation in the bone sense. Additional studies are needed to characterize these released factors and understand their role in differentiation in the bone sense.

To assess collagen presence on the TT EBM and SLM scaffolds, an immunofluorescence assay was performed. The analyses marked COL1A1, the main component of Collagen Type I, and phalloidin. Collagen is the main protein of connective tissues, including bone tissue, while phalloidin marks the main component of the cytoskeleton: F-actin. The immunofluorescence confirmed abundant collagen in both scaffolds, with no significant difference between them. Furthermore, SEM analysis revealed an abundant extracellular bone matrix on the scaffolds cultured in osteogenic medium. The matrix covered the entire scaffold surface and was mineralized, as shown by the EDX analyses at 21 and 40 days.

## 4. Materials and Methods

### 4.1. Trabecular Titanium Scaffolds

Trabecular Titanium scaffolds (Limacorporate Spa, Villanova di San Daniele, Udine, Italy) made of Ti6Al4V were tested in this study. They were produced using two manufacturing technologies (Figure 11): Electron-Beam Melting (EBM) and Selective Laser Melting (SLM). Both scaffold types were fabricated through optimized and validated processes, including post-processing methods, ensuring that both TT structures possess equivalent properties.

Previous studies have demonstrated that trabecular structures produced by both techniques are equivalent in terms of morphology, mechanical properties, and friction characteristics, as described in the introduction and supported by literature data on TT and proprietary technical procedures [4,5,6,7,8,9,12,13].

Therefore, for the purposes of this study, the two scaffold types are considered functionally and structurally comparable, differing only in the manufacturing technology employed.

### 4.2. Isolation of Human Adipose-Derived Stem Cells (hASCs)

This study was conducted in accordance with the 1975 Declaration of Helsinki and approved by the Ethics Committee of the San Matteo Foundation, Research and Care Institute, Pavia, Italy (P-20190023312, 9 April 2019).

The hASCs used in this *in vitro* study were isolated from subcutaneous adipose tissue of healthy donors and obtained during hip replacement surgery. The tissue was thinly chopped and incubated with a digestion buffer (0.01% collagenase type II in DMEM F12-HAM medium, Sigma-Aldrich, Missouri, USA) for 1 h at 37 °C in a shaking water bath.

The suspension was subsequently filtered with a 100 mm filter and centrifuged at 1200 rpm for 10 min at 4 °C. The resulting pellet, containing hASCs, was washed twice with phosphate-buffered saline (PBS, Sigma-Aldrich, St. Louis, MO, USA) and treated with a lysis solution (0.15 M NH_4_Cl, 10 mM KHCO_3_, 0.1 mM Na_2_-EDTA, pH 7.22, Sigma-Aldrich, MO, USA) for 10 min at 4 °C. The hASCs in the pellet were seeded in monolayer at the concentration of 3000/cm^2^ and cultured in growth medium (GM, DMEM F12-HAM supplemented with 100 μg/mL streptomycin, 10% FBS (Fetal Bovine Serum, Dominique DUTSCHER, Bernolshim, France), 100 U/mL penicillin, and 0.25 μg/mL amphotericin, Sigma-Aldrich, MO, USA) in a humidified atmosphere of 95% air with 5% CO_2_ at 37 °C up to 95% confluence.

The adherent cells at 95% confluence were trypsinized with Trypsin Ethylene Diamine Tetra Acetic Acid (EDTA, PAN-biotech, Aidenbach, Germany) and cultured in a new flask [27]. Flow cytometer analysis was performed three times to characterize the markers expressed by the hASCs that were positive for mesenchymal stem cell markers CD73, CD90, and CD105 and negative for hematopoietic cell markers CD34 and CD45 (Navios Beckman Coulter, Indianapolis, IN, USA). The Kaluza 1.2 software package (Beckman Coulter, Brea, USA) was used to acquire, display, and elaborate data. The positive cells were counted and their flowcytometric signal was compared with the corresponding immunoglobulin isotypes controls [28]. 

### 4.3. Cell Seeding and Culture on the Scaffold

To ensure sterile conditions, scaffolds were autoclaved and placed in a 24-well plate.

At the third passage, hASC were trypsinized, and 10,000 cells were seeded onto each scaffold. They were incubated for 1 h at 37 °C with 5% CO_2_ to allow cells’ adhesion on the surface of the scaffolds after GM was added to each scaffold. After a week, an osteogenic differentiation medium (OM, StemProTM Osteogenesis differentiation kit, Thermo Fisher Scientific, Waltham, MA, USA) was added to half of the scaffolds to induce an osteogenic phenotype, while the other half remained in GM.

### 4.4. Conditioned Medium (CM)

To verify the osteoinductive effect of secretome released by cell growth on TT scaffolds, the medium from 3D cultures in the absence of osteogenic factors was collected in both scaffold types. These media (referred to as EBM CM and SLM CM) were used as growth medium for mesenchymal cells cultured in a monolayer.

### 4.5. Cell Seeding on Monolayer

We seeded 10,000 cells on GM in monolayer on a 12-well plate. After one week, the wells were divided into four different conditions: a group continued to grow in GM medium, another group in OM medium, a third group in EBM CM medium, and a fourth group in SLM CM medium. All groups of monolayer cultures were maintained under their respective conditions for 3 weeks.

### 4.6. Proliferation Assay (WST)

To evaluate cell proliferation, optical density was measured at 1, 7, 14, 21, and 28 days after seeding the hASCs on the scaffolds using the WST method (Quick Cell Proliferation Colorimetric Assay Kit, Abcam 155902, Waltham, MA, USA, #K301) according to the manufacturer’s instructions. To ensure the signal was only from cells adhered to the scaffold, the scaffolds were transferred to new wells prior to the assay. This analysis was consistently performed at each culture step.

### 4.7. ALP Activity

We measured ALP activity at 7, 14, and 21 days after the seeding to confirm osteogenic differentiation. For cell growth on scaffolds, the constructs were moved to new wells before the ALP assay was carried out to exclude the signal arising from cells adhering to the plastic plates, while for cells on monolayer, ALP was conducted directly in the well. The samples were washed in PBS and incubated with 5 mM p-nitrophenyl-phosphate in 50 mM glycine, 1 mM ZnSO_4_, and 1 mM MgSO_4_ (Sigma-Aldrich, MO, USA) with a pH of 10.5 for 10 min at 37 °C. P-nitrophenol presented an absorbance of 410 nm. The ALP activity was expressed as the formation of 1 nmol p-nitrophenol/100 K cells (Sigma-Aldrich, MO, USA). We calculated the cell number with the WST test, and then we performed the ALP activity assay.

### 4.8. RNA Isolation and Reverse Transcriptase Quantitative Real-Time PCR (qRT-PCR)

To study the osteogenic cell differentiation, we tested the same typical gene of osteoblast phenotype. After 21 days from differentiation, RNA was extracted from the scaffolds and the monolayer with the Gene MATRIX KIT for RNA purification (Biosigma, Cona, Italy) to evaluate gene expression. Using random hexamers and M-MLV Reverse Transcriptase (Promega, Milano, Italy), the total RNA extracted was reverse-transcribed into cDNA. Quantitative real-time PCR (qRT-PCR) was performed in triplicate using 2 μL cDNA, using specific primers from Qiagen (Qiagen, Hilden, Germany) (Table 1): *COL1A1* (Collagen alpha-1-chain gene), *ALP* (alkaline phosphatase gene), *OSX* (Transcription factor Sp7), and *BGLAP* (Osteocalcin gene). qPCR was performed using Quantifast-SYBR Green PCR Kit (Qiagen, Hilden, Germany) and StepOnePlusTM Real-Time PCR System (Applied BiosystemsTM, Thermo Fisher Scientific, Waltham, MA, USA), according to the manufacturer’s instructions. To normalize the qPCR reactions, the housekeeping gene *β2M* expression (beta-2 microglobulin gene, Qiagen, Hilden, Germany) was used. After the PCR run, the melting curves were generated to identify the melting temperatures of specific products. The gene expression results of differentiated cells were expressed as fold change versus expression of hASC after 28 days of culture in the growth medium (control).

### 4.9. Immunofluorescence Assay

We conducted an immunofluorescence assay to highlight the presence of extracellular matrix on the scaffold, marking phalloidin and COL1A1. We conducted the analysis on EBM and SLM scaffolds cultured in OM. After 21 days from differentiation, the cells grown on the scaffolds were fixed with paraformaldehyde (Sigma-Aldrich, MO, USA) at 4% (PFA 4%) in PBS for 30 min. Cells were permeabilized with TRITON-X (Sigma-Aldrich, MO, USA) 0.4%, washed three times with PBS 1X, and incubated with the diluted (1:50) primary antibody against COL1A1 (PA1-26204, Thermo Fisher Scientific, Waltham, MA, USA) overnight at 4 °C. After, the scaffolds were washed three times with PBS 1X and incubated for 30 min at RT with a secondary antibody—diluted 1:1000 (A-21207, Thermo Fisher Scientific, Waltham, MA, USA). The scaffolds were washed three times with PBS 1X and incubated with PHALLOIDIN (PHALLOIDIN Atto 488 Sigma, Aizu, Japan). Finally, the scaffolds were mounted with an anti-fading mounting solution (ProLongTM Gold Antifade Mountant Thermo Fisher Scientific, Waltham, MA, USA) and kept at 4 °C until their visualization with a fluorescence microscope (Nikon Eclipse 80i (Nikon, Tokyo, Japan)).

### 4.10. Scanning Electron Microscope (SEM) Analysis

SEM analysis was conducted to evaluate the extracellular matrix. After 21 and 40 days from differentiation, the scaffolds were washed with PBS and then fixed for 2 h with glutaraldehyde (Sigma-Aldrich, Missouri, USA) 3%. The scaffolds were dehydrated with different ethanol grades, from 50%, 70%, 90%, to 100%. A high-resolution SEM (EVO 40 SMART scanning electron microscope, Zeiss and TESCAN Mira 3 XMU, Pavia, Italy) was employed at 20 kV to perform the microstructural characterization. We conducted two different analyses: BSE (backscattered electron) and SE (scattered electron). BSE uses the interaction between atomic nuclei and the electrons of the analyzed sample to create a grayscale image, while SE uses secondary electrons emitted by the affected material to obtain a high-resolution three-dimensional image. The scaffolds were coated with carbon, using a Cressington carbon coater 208c (Cressington Scientific Instruments, Watford, UK) before observing. SEM analyses were conducted at Arvedi Laboratory, CISRiC (Centro Interdipartimentale di Studi e Ricerche per la Conservazione del Patrimonio Culturale), University of Pavia, Pavia, Italy.

### 4.11. Statistical Analysis

All data are expressed as mean ± SEM (standard error of the mean). Results were analyzed using multiple or unpaired *t*-tests, or one-way ANOVA, with a significance threshold set at *p*-value < 0.05. Statistical analyses were performed using GraphPad Prism 8.4.2. (GraphPad Software, La Jolla, San Diego, CA, USA).

## 5. Conclusions

This study demonstrates that Trabecular Titanium is able to induce adhesion, proliferation, and differentiation of mesenchymal stem cells into bone, regardless of the manufacturing methodology.

The studies conducted to date indicate that the micro- and macro-structure features are responsible for the osteoinductive properties. However, further studies are needed (e.g., the analysis of the secretome or the analysis of the proteins released by hASCs seeded on scaffolds) to deepen our knowledge of the mechanism of the osteoinductive capacity of these scaffolds.

## Figures and Tables

**Figure 1 ijms-26-06354-f001:**
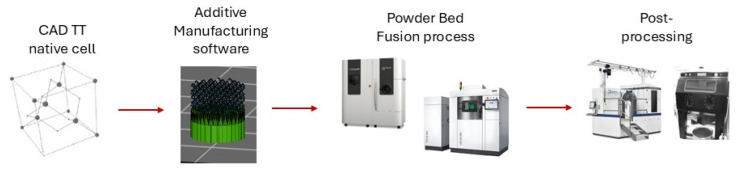
Schematic illustration of the main TT fabrication steps.

**Figure 2 ijms-26-06354-f002:**
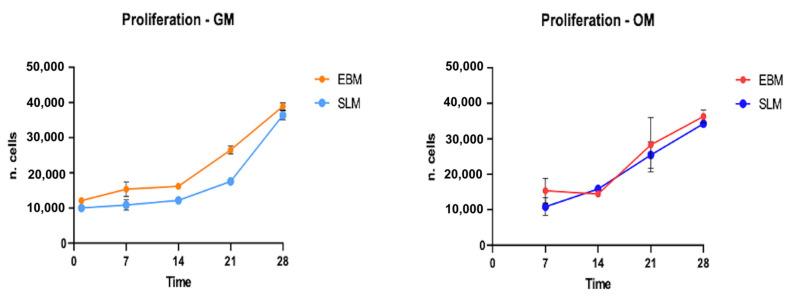
The graphic represents the hASC growth curve at 1, 7, 14, 21, and 28 days in growth medium (GM) on TT EBM scaffolds (orange line) and TT SLM scaffolds (light blue line) and in osteogenic differentiation medium (OM) on TT EBM scaffolds (red line) and on TT SLM scaffolds (blue line). Cell differentiation starts from 7 days after seeding. Statistical analysis was conducted using a *t*-test for unpaired data (GraphPad Prism 8.4.2).

**Figure 3 ijms-26-06354-f003:**
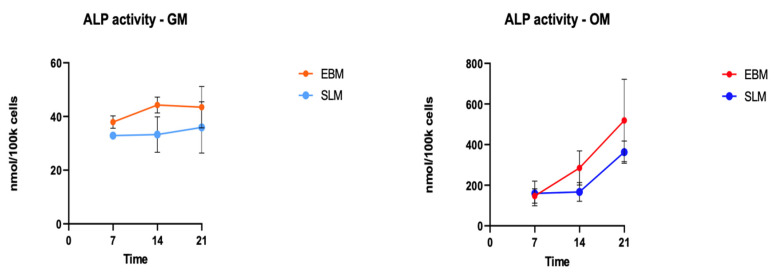
ALP activity of hASCs at 7, 14, and 21 days of culture in growth medium (GM) and osteogenic differentiation medium (OM). The graph shows the following conditions: cells grown on TT EBM scaffold in GM medium (orange), cells grown on TT SLM scaffold in GM medium (light blue), grown on TT EBM scaffold in OM medium (red), and cells grown on TT SLM scaffold in OM medium (blue). Statistical analysis was conducted using a *t*-test for unpaired data (GraphPad Prism 8.4.2).

**Figure 4 ijms-26-06354-f004:**
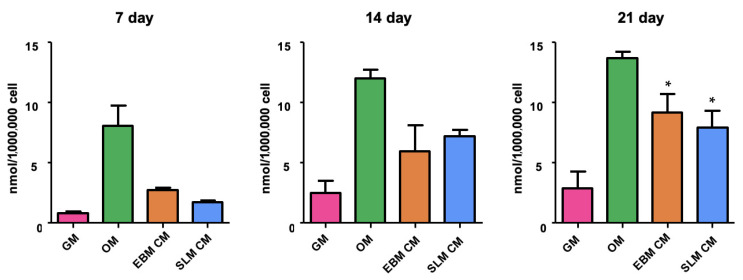
ALP activity of hASCs cultured for 7, 14, and 21 days in growth medium (GM), conditioned medium (CM, GM + SLM, and GM + EBM), and osteogenic differentiation medium (OM). Statistical analysis was performed using one-way ANOVA (GraphPad Prism 8.4.2). Significance: * (*p* < 0.05 vs. GM).

**Figure 5 ijms-26-06354-f005:**
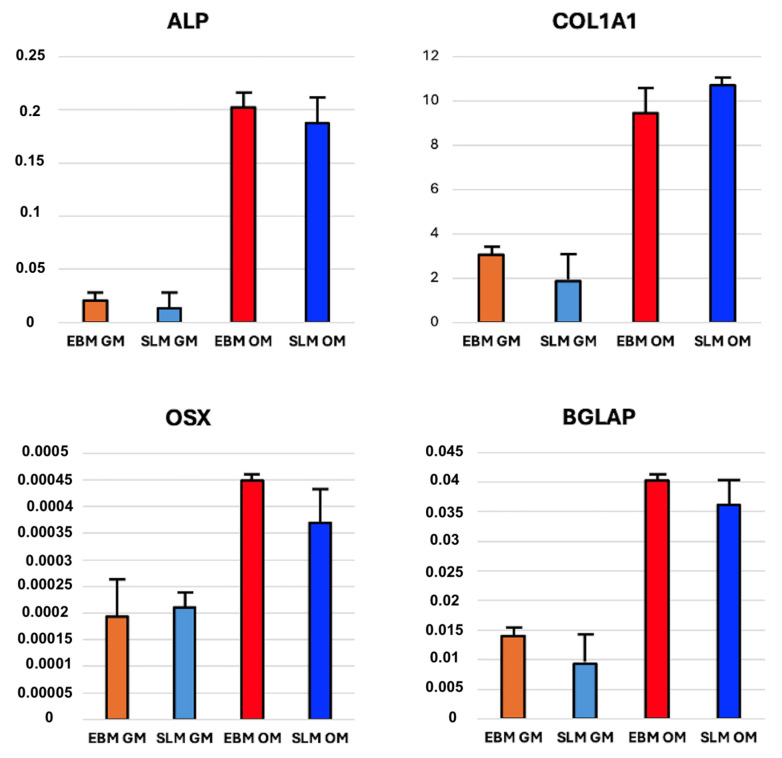
Gene expression of hASCs seeded on TT EBM and TT SLM scaffolds and cultured in growth medium (GM) or osteogenic medium (OM) for 21 days. Statistical analysis was conducted using a *t*-test for unpaired data (GraphPad Prism 8.4.2).

**Figure 6 ijms-26-06354-f006:**
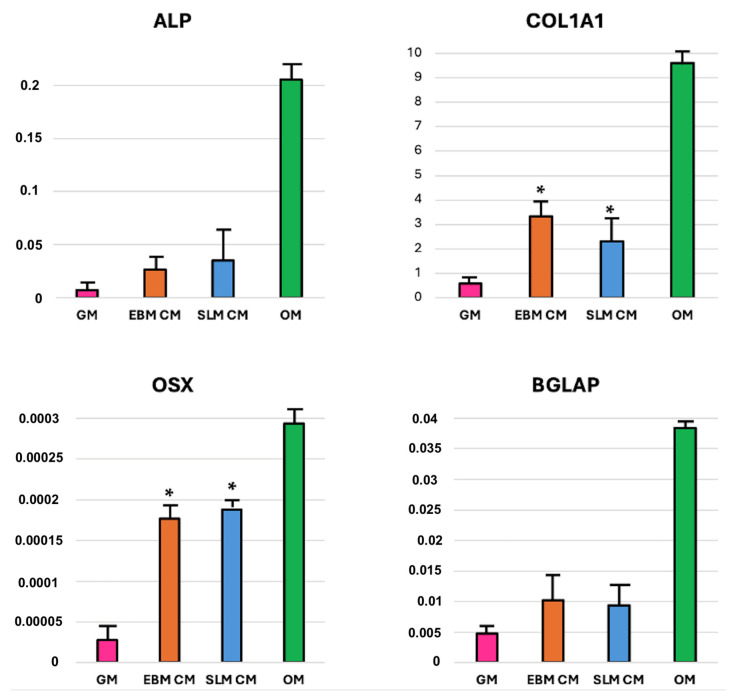
Gene expression of the hASCs seeded on a monolayer and cultured in growth medium (GM), osteogenic differentiation medium (OM), EBM conditioned medium (CM), and SLM conditioned medium (CM) for 21 days. Statistical analysis conducted with one-way ANOVA (GraphPad Prism 8.4.2). Significance: * (*p* < 0.05 vs. GM).

**Figure 7 ijms-26-06354-f007:**
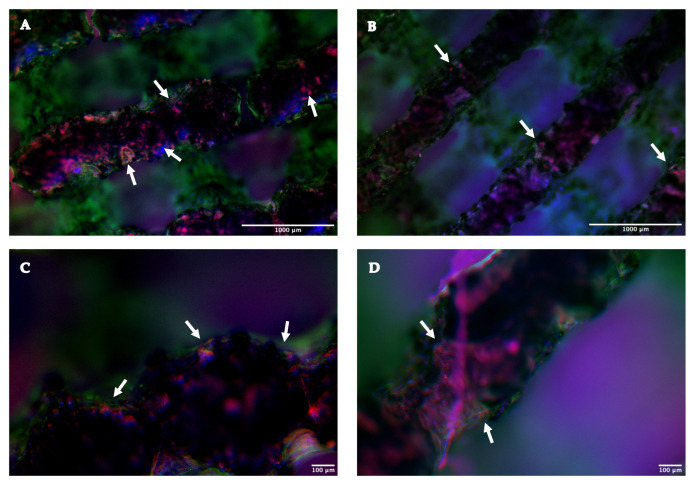
Immunofluorescence assay at 28 days from the cell seeding: (**A**) COL1A1 and PHALLOIDIN on TT EBM scaffold, mag 4×; (**B**) COL1A1 and PHALLOIDIN on TT SLM scaffold, mag 4×; (**C**) COL1A1 and PHALLOIDIN on TT EBM scaffold, mag 10×; (**D**) COL1A1 and PHALLOIDIN on TT SLM scaffold, mag 10×. In the image, the cell nucleus (DAPI) is marked in blue, the cytoskeleton (phalloidin) is marked in green, while the collagen (COL1A1) is marked in red and highlighted with white arrow.

**Figure 8 ijms-26-06354-f008:**
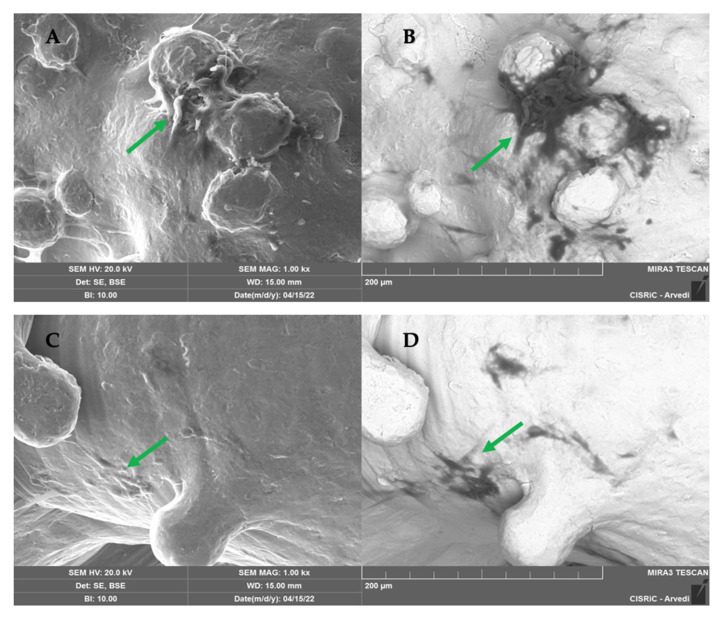
SEM images of TT EBM (**A**,**B**) and TT SLM (**C**,**D**) scaffolds cultured in growth medium (GM) at 28 days from the cell seeding, captured using the secondary electron detector (SE) and the backscattered electron detector (BSE). In BSE images (**B**,**D**), cells and the matrix appeared dark, while the scaffolds appeared white. The green arrow highlights matrix deposition. Mag.: 1.

**Figure 9 ijms-26-06354-f009:**
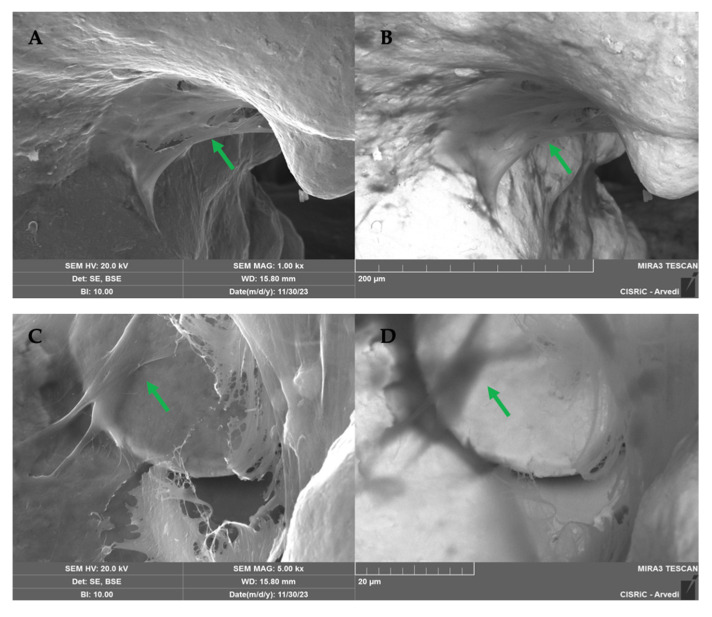
SEM images of SLM (**A**,**B**) and EBM (**C**,**D**) scaffolds cultured in osteogenic differentiation medium (OM) at 28 days from the cell seeding, taken with the scattered electron detector (SE) and the backscattered electron detector (BSE). With the BSE detector (**B**,**D**), cells and matrix appeared dark, and scaffolds appeared white. The green arrow highlights the matrix deposition. Mag.: 1 k×.

**Figure 10 ijms-26-06354-f010:**
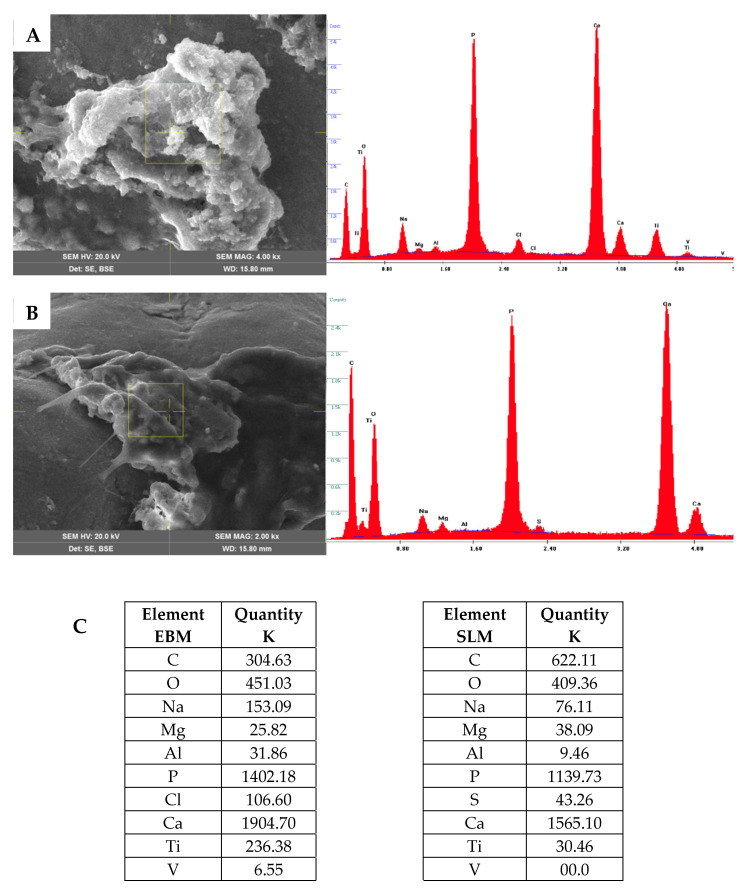
SEM images of TT EBM (**A**) and TT SLM (**B**) scaffolds cultured in osteogenic differentiation medium (OM) at 28 days from the cell seeding. On the left, the image shows the scaffold magnification obtained with the secondary electron detector (SE), while on the right, the corresponding EDS analysis is presented. (**C**) Tables with element quantification, as seen in the SEM analysis.

**Figure 11 ijms-26-06354-f011:**
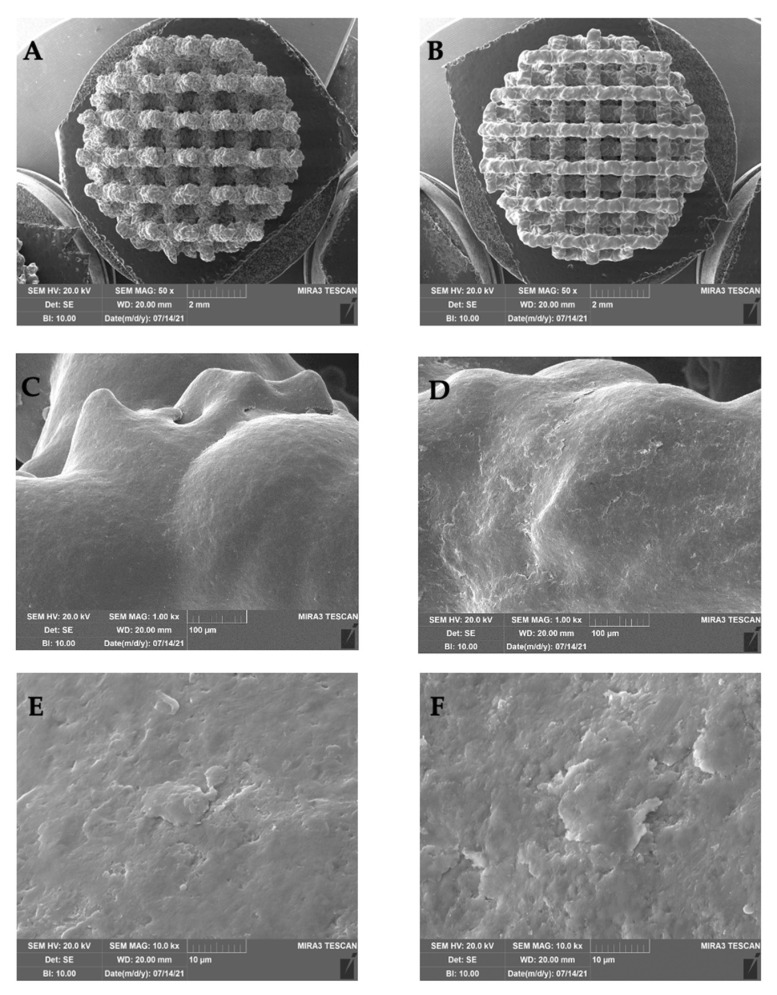
Scanning electron microscopy (SEM) study of the utilized scaffolds. Panel (**A**) indicates EBM scaffold, while panel (**B**) indicates SLM scaffold (mag. 50×). Panels (**C**,**D**) show a higher magnification of the surface roughness (mag. 1.00 k×). Panels (**E**,**F**) show a higher magnification of the surface roughness (mag. 10.0 k×).

**Table 1 ijms-26-06354-t001:** Primers used in real-time PCR experiments.

Gene	Target Transcript	Quantitect Primer Assay (Quiagen)	Amplicon Length
*ALP*	NM_000478	QT00012957	110 bp
*COL1A1*	NM_000088	QT00037793	127 bp
*OSX*	NM_152860	QT00213514	120 pb
*BGLAP*	NM_199173	QT00232771	90 pb
*β2M*	NM_004048	QT00088935	98 pb

## Data Availability

Data are contained within the article.

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
