# Peer review of "Trabecular Titanium Architecture Drives Human Mesenchymal Stem Cell Proliferation and Bone Differentiation"

_ijms, 2025, doi:10.3390/ijms26136354_

Round 1
Reviewer 1 Report
Comments and Suggestions for Authors
The manuscript "Trabecular Titanium architecture drives human mesenchymal
stem cells proliferation and bone differentiation" showes the comparison of the two methods for the preparation of Ti scaffolds.
In is interesting for the readers but it needs the improvement.
The comments:
In Results
The abbreviation CM, GM and OM should be explained in results and figure legends as I tis mentioned on page 13.
Did the different methods used for the preparation of samples lead to different hydrophilicity of the samples? Was it already measured although on plane samples?
In Fig. 5 there is only mentioned a significance between EBM CM, SLM CM, but not between OM – it looks there is a significant difference, however it is not marked on the figures. Was this group added into the statistical evaluation? If yes, please, indicate the significant difference for this group as well. The increase of number of tested groups may significantly change the level of significance and the OM group should be included the statistical evaluation.
In Fig. 5 there is mentioned only day 21. It is not optimum date for early markers, such as ALP, or medium-term marker, such as collagen. Could you add the information from the earlier time interval, e.g. Day 7, please?
In Figure legends (Fig. 6, 7, 8), could the authors mention the time of culture of the cells on the scaffolds?
In 4.9. It is mentioned that two time-intervals were used for SEM – 21 and 40 days from differentiation, however, only one time interval is showed on the Fig 7 and 8. Please, modify the figure legends and the text in “SEM analysis“, page 8.
In Fig. 9., on the right part, the letter are not readable, could the authors prepare the table with the elements and values?
In Discussion there is not enough examples comparing both methodologies. Could the authors discuss and compare also other Ti alloys where these methods are used? Or any similar materials?
Could they discuss also biomechanical properties of both samples?
The referenced 18-20 did not support the previous sentence in which there is mentioned that Osx inhibits Bglap and Col1a1. I did not find any relevant article supporting such theory. Please, could you modify it?
The sentence on lines 274-276 is not finished.
In Methods:
From the Figure 10 it seems that the roughness of TT EBS may be higher compared to TT SLM. Did you measure it? Could the author add this information or citation?
In 4.2. concentration of 3000 should be changed to “….3,000“
In 4.6. WST kit refers to proliferation, viability and metabolic activity but not directly to Adhesion. 4.6. name should be modified. Also Abcam number - probably ab155902 or another one should be mentioned.
In 4.7 The authors calculated ALP activity as a ratio of the value per 100 k cells. Where did they find the cell number? From the WST assay – from different wells? Please, explain.
References:
Citation 11 is not full, please, complete it.
Author Response
We thank the reviewer for the valuable advice, we have made the changes has requested by you.
- The abbreviation CM, GM and OM should be explained in results and figure legends as I tis mentioned on page 13.
The abbreviation GM and OM has been correctly reported on lines 104-105, while we thank for advice and add the explanation for the abbreviation CM. We add the complete name to all figures.
- Did the different methods used for the preparation of samples lead to different hydrophilicity of the samples? Was it already measured although on plane samples?
We thank the reviewer for this insightful comment. We believe that, since the scaffolds demonstrated comparable biological performance—particularly in terms of cell adhesion and proliferation—additional surface characterization, such as hydrophilicity measurements, was not deemed necessary.
- In Fig. 6 there is only mentioned a significance between EBM CM, SLM CM, but not between OM – it looks there is a significant difference, however it is not marked on the figures. Was this group added into the statistical evaluation? If yes, please, indicate the significant difference for this group as well. The increase of number of tested groups may significantly change the level of significance and the OM group should be included the statistical evaluation.
We did not compare the results obtained from the gene expression of cells conditioned with CM compared to OM, because we expected significant results with OM, given that it contains bone differentiation factors. The aim of the experiment was to evaluate whether the cells grown on the scaffolds released factors into the medium that were able to induce bone differentiation without the addition of external factors. Therefore, we wanted to compare the results obtained with CM compared to growth medium.
- In Fig. 6 there is mentioned only day 21. It is not optimum date for early markers, such as ALP, or medium-term marker, such as collagen. Could you add the information from the earlier time interval, e.g. Day 7, please?
We thank the reviewer for the comment, we evaluated gene expression in an interval earlier than 21 days but we did not find significant changes in cells treated in CM compared to GM. Since CM does not contain external factors that induce bone differentiation, but are the factors released by cells seeded on scaffolds, it was necessary to extend the exposure times of the cells in CM.
- In Figure legends (Fig. 7, 8,9), could the authors mention the time of culture of the cells on the scaffolds?
Yes, we do it.
- In 4.9. It is mentioned that two time-intervals were used for SEM – 21 and 40 days from differentiation, however, only one time interval is showed on the Fig 7 and 8. Please, modify the figure legends and the text in “SEM analysis“, page 8.
The text only reports the results of the analysis performed at 21 days after bone differentiation; the analyses conducted at 40 days have not been reported in the text because they confirmed the results obtained at 21 days, as reported on line 237-238.
- In Fig. 10., on the right part, the letter are not readable, could the authors prepare the table with the elements and values?
Yes, we do it.
- In Discussion there is not enough examples comparing both methodologies. Could the authors discuss and compare also other Ti alloys where these methods are used? Or any similar materials?
We thank the reviewer for the comment, we add it at line 300-309.
- Could they discuss also biomechanical properties of both samples?
Additional details regarding the biomechanical properties of the tested scaffolds, which inherit the characteristics of the lattice structure called Trabecular Titanium, have been added to the Introduction (lines 50-64) and to the Materials and Methods section (lines 353- 363).
- The referenced 18-20 did not support the previous sentence in which there is mentioned that Osx inhibits Bglap and Col1a1. I did not find any relevant article supporting such theory. Please, could you modify it?
The 18 referenced is correct, we change the other with 2 more appropriate references.
- The sentence on lines 347-348 is not finished.
We correct it.
- From the Figure 11 it seems that the roughness of TT EBS may be higher compared to TT SLM. Did you measure it? Could the author add this information or citation?
We thank the reviewer for this helpful comment. Additional details to better address the uniformity of the tested scaffold have been added in the paper (lines 353- 363).
Despite minimal morphological differences may be observed between the two reported images, previous studies have shown that, under the chosen production conditions, Trabecular Titanium manufactured using EBM and SLM technologies exhibits equivalent properties. The manufacturing processes and parameters are subject to continuous validation and monitoring through internal procedures established within the company’s Quality Management System. This ensures consistent and reproducible lattice characteristics across batches, including the specimens analyzed in this study.
- In 4.2. concentration of 3000 should be changed to “….3,000“
We correct it.
- In 4.6. WST kit refers to proliferation, viability and metabolic activity but not directly to Adhesion. 4.6. name should be modified. Also Abcam number - probably ab155902 or another one should be mentioned.
We correct it.
- In 4.7 The authors calculated ALP activity as a ratio of the value per 100 k cells. Where did they find the cell number? From the WST assay – from different wells? Please, explain.
We calculated the cell number with the WST test, first performed the WST test and then performed the ALP activity assay using the same wells. This is possible because the WST test is a viability test.
- Citation 11 is not full, please, complete it
We correct it.
Reviewer 2 Report
Comments and Suggestions for Authors
The manuscript entitled „Trabecular Titanium architecture drives human mesenchymal stem cells proliferation and bone differentiation“ by Caliogna et al. is of some interest for the readers of IJMS.
The authors analyzed the effects of two different manufacturing methods for titanium-based trabecular scaffolds on the cytocompatibility of human adipose-derived mesenchymal stem cells (MSCs). The two fabrication techniques compared were electron beam melting (EBM) and the more recent approach of selective laser melting (SLM).
MSCs exhibited comparable growth on both scaffold types. Osteogenic differentiation, induced by osteogenic differentiation medium, was also similar between the two scaffold groups. Medium conditioned by MSCs cultured on the test scaffolds exhibited some osteoinductive capacity; however, the level of osteoinduction did not reach that achieved by the dedicated osteogenic medium.
Additional immunofluorescence staining demonstrated collagen deposition on the scaffold surfaces by the MSCs in both groups. Energy-dispersive X-ray spectroscopy (EDS) analysis further revealed that this collagenous matrix was calcified on both scaffold types.
An original study is presented, as verified through PubMed and Google Scholar. The general conclusion is that both manufacturing methods of trabecular titanium scaffolds resulted in comparable cytocompatibility. Consequently, the more cost-effective or technically favorable method may be selected for scaffold production. The scientific merit of this study is at least moderate, as it contributes to the growing body of knowledge required for the advancement of promising scaffold designs for bone regeneration.
Comments:
- Please provide additional fundamental details regarding the tested scaffolds. Do not refer exclusively to previously published studies. For instance, what are the pore sizes, and what is the height or depth of surface irregularities (e.g., "peaks" and "pits")?
- Based on the images shown in Figure 10, EBM and SLM-manufactured scaffolds appear to differ. The EBM scaffold surface seems more uniform, whereas SLM-fabricated scaffolds display bordered pits. Could this affect the comparability of the results?
- Is it possible to quantify the surface roughness of the scaffolds?
- ALP activity: Figures 2 and 3 appear to show the same dataset. If this is the case, consider removing Figure 2 and retaining only Figure 3 to avoid redundancy.
- Figure 6: Please provide a scale bar for the immunofluorescence images.
- Do you have any hypothesis about which factor(s) present in the conditioned medium may have been responsible for the observed osteoinductive effect? Please expand on this in the discussion.
- The discussion section appears underdeveloped and lacks sufficient depth. Please address point 6 and include a clear conclusion as well as the limitations of the study.
Author Response
We thank the reviewer for the valuable advice, we have made the changes as requested by you.
- Please provide additional fundamental details regarding the tested scaffolds. Do not refer exclusively to previously published studies. For instance, what are the pore sizes, and what is the height or depth of surface irregularities (e.g., "peaks" and "pits")?
Following the reviewer’s suggestion, additional details regarding the tested scaffolds, which inherit the characteristics of the lattice structure called Trabecular Titanium, have been added to the Introduction (lines 50-64) and to the Materials and Methods sections (lines 353-363).
- Based on the images shown in Figure 10, EBM and SLM-manufactured scaffolds appear to differ. The EBM scaffold surface seems more uniform, whereas SLM-fabricated scaffolds display bordered pits. Could this affect the comparability of the results?
We thank the reviewer for this helpful comment. Additional details to better address the uniformity of the tested scaffold have been added to the paper (lines 353-363). For completeness, it is worth noting that although minimal morphological differences may be observed between the two sample sets, previous studies have shown that, under these production conditions, Trabecular Titanium manufactured using the two technologies exhibits equivalent properties. The manufacturing processes and parameters are subject to continuous validation and monitoring through internal procedures established within the company’s Quality Management System. This ensures consistent and reproducible lattice characteristics across batches, including the specimens analyzed in this study.
- Is it possible to quantify the surface roughness of the scaffolds?
We believe that, since friction depends not only on surface roughness but also on other factors such as pore shape and size, the friction coefficient serves as a more comprehensive indicator for globally assessing the properties of lattices produced by additive manufacturing technologies. As mentioned in the introduction (lines 50-64), a dedicated study evaluating the frictional properties of Trabecular Titanium fabricated via EBM and SLM is available. Because the scaffold used in the present study fully inherits these characteristics, surface roughness was not directly measured here.
- ALP activity: Figures 3 and 4 appear to show the same dataset. If this is the case, consider removing Figure 2 and retaining only Figure 3 to avoid redundancy.
The authors thank the reviewer for his advice. We would like to clarify that figure 3 refers only to the ALP activity of cells cultured on scaffold in growth medium (left graph) and the ALP activity of cells cultured on scaffold in osteogenic medium (right graph), while figure 4 refers to the ALP activity of cells seeded in monolayer. The data show the ALP activity of cells cultured in monolayer in three different conditions: growth medium, bone differentiation medium and conditioned medium. The conditioned medium derived from cells seeded in growth medium on EBM and SLM scaffold. The aim of the experiment was to show that cells seeded on scaffold release in culture medium many different factors, which can induce bone differentiation without osteogenic factors.
- Figure 7: Please provide a scale bar for the immunofluorescence images.
We added the scale bar to the images.
- Do you have any hypothesis about which factor(s) present in the conditioned medium may have been responsible for the observed osteoinductive effect? Please expand on this in the discussion.
In order to clarify the reason for inducing bone differentiation by the factors released into the medium, it would be necessary to analyze both the secretome, which could contain several different factors able to induce differentiation both the protein component that the cells can release during culture. Probably an additional stimulus provided by the three-dimensional structure of the scaffold, which mimics the trabecular bone. We expand the discussion on this topic by following your advice.
- The discussion section appears underdeveloped and lacks sufficient depth. Please address point 6 and include a clear conclusion as well as the limitations of the study.
We thank the reviewer for the advice and expanded the discussion section (line 336-342).
Reviewer 3 Report
Comments and Suggestions for Authors
Dear Authors,
Thank you for your interesting and valuable contribution. I believe your work holds significant potential for applications in bone tissue engineering. Microstructured scaffolds—beyond just titanium—play a crucial role in the integration of bone implants. Your study is methodologically sound, with an adequate sample size and appropriate analytical techniques. I would recommend this manuscript for publication following minor revisions and additions.
The introduction, while concise, effectively outlines the topic. However, for readers who may not be fully familiar with the material preparation process, it would be beneficial to include a schematic or illustration of the TT carrier fabrication. Additionally, a brief comparison of different preparation methods could enhance the context.
Regarding the references, the manuscript would benefit from the inclusion of more recent literature (preferably from the past five years) to better reflect the current state of research in this field.
I also have a few reservations regarding the description of the methods and the presentation of the results, which I believe could be clarified or expanded upon:
Section 2.1 – Adhesion and Proliferation
While the proliferation data (based on cell number) is clearly presented and appropriate, the adhesion aspect is less clear. Is the 24-hour time point intended to reflect adhesion? If so, it would be helpful to explicitly state this. Additionally, did you perform any specific assays or imaging to assess how cells attach to the substrate (e.g., quantification of adhered cells, morphology analysis, or focal adhesion markers)?
Section 2.1 – Immunofluorescence Staining and Figure 6
I understand the challenges posed by the complex 3D structure in terms of imaging and focus. You mention using a widefield fluorescence microscope—have you considered Z-stack imaging with deconvolution or extended depth of field processing to improve image clarity and depth representation? Also, could you clarify why cell nuclei staining was not included? Including simple DAPI/Hoechst or a similar stain would help in assessing cell distribution and morphology.
Section 4.1 – TT Scaffolds
You mention that both substrates have similar mechanical and frictional properties, citing references from 2010–2015. Given the apparent differences in microstructure shown in Figure 10E and 10F, have you validated these properties experimentally in your current study? For example, did you measure surface roughness (e.g., Ra values) or perform like AFM testing to confirm these similarities?
Section 4.2 – Cell Source
Out of curiosity, why did you choose adipose-derived stem/stromal cells (ASCs) over bone marrow-derived mesenchymal stem cells (MSCs)? Many studies prefer MSCs due to their reportedly great osteogenic potential. That said, we also use ASCs (both human and porcine) in our work because they are easier to harvest and have demonstrated the ability to differentiate into osteoblast-like cells.
Section 4.2 – Trypsin-EDTA
Just a minor note: the use of Trypsin-EDTA (line 319) is standard and widely accepted.
Section 4.3 – Differentiation Medium
I’m also curious about the composition of the osteogenic differentiation medium. Could you specify which supplements were added to induce osteogenesis (e.g., dexamethasone, β-glycerophosphate, ascorbic acid)? This would be helpful for reproducibility and comparison with other studies.
Author Response
We thank the reviewer for the valuable advice, we have made the changes as requested by you
- Section 2.1 – Adhesion and Proliferation
While the proliferation data (based on cell number) is clearly presented and appropriate, the adhesion aspect is less clear. Is the 24-hour time point intended to reflect adhesion? If so, it would be helpful to explicitly state this. Additionally, did you perform any specific assays or imaging to assess how cells attach to the substrate (e.g., quantification of adhered cells, morphology analysis, or focal adhesion markers)?
We thank the reviewer for the advice and explained better in line 101-102 the adhesion aspect. For the evaluation of the adhesion, we performed the WST test that allows evaluating the quantification of cell proliferation and cell viability. The colorimetric assay is based on the cleavage of WST-1 (a tetrazolium salt) to formazan by mitochondrial dehydrogenases in living cells. The formazan production is proportional to the number of live and active cells.
- Section 2.1 – Immunofluorescence Staining and Figure 6
I understand the challenges posed by the complex 3D structure in terms of imaging and focus. You mention using a widefield fluorescence microscope—have you considered Z-stack imaging with deconvolution or extended depth of field processing to improve image clarity and depth representation? Also, could you clarify why cell nuclei staining was not included? Including simple DAPI/Hoechst or a similar stain would help in assessing cell distribution and morphology.
We tried to acquire the images using a confocal microscope, but the scaffold structure does not allow clear images because the metal alloy of the scaffold is autofluorescent. We removed the cells nuclei stained with DAPI from the image because it made images not clear preventing the collagen signal from being clearly visible. Because it has been asked, we will insert those in the text.
3. Section 4.1 – TT Scaffolds
You mention that both substrates have similar mechanical and frictional properties, citing references from 2010–2015. Given the apparent differences in microstructure shown in Figure 10E and 10F, have you validated these properties experimentally in your current study? For example, did you measure surface roughness (e.g., Ra values) or perform like AFM testing to confirm these similarities?
To clarify, additional details regarding the tested scaffolds, which inherit the characteristics of the Trabecular Titanium lattice structure, have been added to the Introduction (lines 50-64) and to the Materials and Methods sections (lines 353-363).
Despite minimal morphological differences may be observed between the two reported images, previous studies have shown that, under these production conditions, Trabecular Titanium manufactured using EBM and SLM technologies exhibits equivalent properties. The manufacturing processes and parameters are subject to continuous validation and monitoring through internal procedures established within the company’s Quality Management System. This ensures consistent and reproducible lattice characteristics across batches, including the specimens analyzed in this study.
We believe that since friction depends not only on surface roughness but also on other factors such as pore shape and size, the friction coefficient serves as a more comprehensive indicator for globally assessing the properties of lattices produced by additive manufacturing technologies. As the scaffold used in the present study fully inherits these characteristics, surface roughness was not directly measured here. The dedicated study is included in the same paragraphs (Salatin et al, 2022, ISTA - Preliminary Comparison between Trabecular Titanium Porous Structure Manufactured via EBM and SLM Technologies Friction and Primary Stability Performances)
4. Section 4.2 – Cell Source
Out of curiosity, why did you choose adipose-derived stem/stromal cells (ASCs) over bone marrow-derived mesenchymal stem cells (MSCs)? Many studies prefer MSCs due to their reportedly great osteogenic potential. That said, we also use ASCs (both human and porcine) in our work because they are easier to harvest and have demonstrated the ability to differentiate into osteoblast-like cells.
We have chosen ASCs because they are easy to find without causing major morbidity to the patient. In addition, many cells can be isolated from a small amount of tissue. They still have a good osteogenic differentiation potential. In addition, our laboratory collaborates with an orthopedic clinic where adipose tissue is widely available as waste tissues following prosthetic surgeries. this reason, adipose tissue an important cells’ source for testing new biomaterials.
5. Section 4.2 – Trypsin-EDTA
Just a minor note: the use of Trypsin-EDTA (line 390) is standard and widely accepted.
The citation 27 refers to the procedure used (cells detached at 95% confluence) and not to the type of reagent used to detach them.
6. Section 4.3 – Differentiation Medium
I’m also curious about the composition of the osteogenic differentiation medium. Could you specify which supplements were added to induce osteogenesis (e.g., dexamethasone, β-glycerophosphate, ascorbic acid)? This would be helpful for reproducibility and comparison with other studies.
The culture medium we used is a ready-to-use one (Thermo put code) where the precise composition of the factors that induce differentiation is under patent.
Round 2
Reviewer 1 Report
Comments and Suggestions for Authors
The authors modified the manuscript according to the comments, and clarified some questions, however, the procedure of ALP measurement was explained only in the Author response but not in a manuscript. To understand the article properly, it should be added in Methods (4.7.) (point 15 of the comments).
Author Response
We thank the reviewer for the valuable advice, we have made the changes has requested by you.
- The authors modified the manuscript according to the comments, and clarified some questions, however, the procedure of ALP measurement was explained only in the Author response but not in a manuscript. To understand the article properly, it should be added in Methods (4.7.) (point 15 of the comments).
we do it
Reviewer 2 Report
Comments and Suggestions for Authors
All my questions and issues were answered.
Author Response
We thank the author for the advice and the answer